# RANet: Relationship Attention for Hyperspectral Anomaly Detection

**Yingzhao Shao [1], Yunsong Li [1,*], Li Li [2], Yuanle Wang [2,3], Yuchen Yang [2], Yueli Ding [2], Mingming Zhang [2], Yang Liu [2] and Xiangqiang Gao [2]**

[1] State Key Laboratory of Integrated Services Networks, Xidian University, Xi'an 710071, China; shaoyz@cast504.com

[2] China Academy of Space Technology (Xi'an), Xi'an 710100, China; li_li@castxian.com (L.L.); wang.y.l@stu.xjtu.edu.cn (Y.W.); yangyc75@cast504.com (Y.Y.); dingyl@cast504.com (Y.D.); zhangmingming@cast504.com (M.Z.); liuy4@cast504.com (Y.L.); xggao@buaa.edu.cn (X.G.)

[3] School of Microelectronics, Xi'an Jiaotong University, Xi'an 710049, China

[*] Correspondence: ysli@mail.xidian.edu.cn

**Abstract:** Hyperspectral anomaly detection (HAD) is of great interest for unknown exploration. Existing methods only focus on local similarity, which may show limitations in detection performance. To cope with this problem, we propose a relationship attention-guided unsupervised learning with convolutional autoencoders (CAEs) for HAD, called RANet. First, instead of only focusing on the local similarity, RANet, for the first time, pays attention to topological similarity by leveraging the graph attention network (GAT) to capture deep topological relationships embedded in a customized incidence matrix from absolutely unlabeled data mixed with anomalies. Notably, the attention intensity of GAT is self-adaptively controlled by adjacency reconstruction ability, which can effectively reduce human intervention. Next, we adopt an unsupervised CAE to jointly learn with the topological relationship attention to achieve satisfactory model performance. Finally, on the basis of background reconstruction, we detect anomalies by the reconstruction error. Extensive experiments on hyperspectral images (HSIs) demonstrate that our proposed RANet outperforms existing fully unsupervised methods.

**Keywords:** anomaly detection; hyperspectral image; graph attention network (GAT); convolutional autoencoder (CAE)



## 1. Introduction

As an image type that can capture spectral intrinsic information with the help of advanced imaging technology, hyperspectral images (HSIs) are usually represented in three dimensions; the first two dimensions show spatial information, and the third dimension records spectral information, which is used to reflect different properties of different substances [1–6]. Compared to ordinary RGB images, HSIs have more spectral bands, which can store richer information and depict more details of the captured scenes. Currently, HSIs have been widely used in the fields of resource exploration, environmental monitoring, precision agriculture and management [7–11]. On this basis, various data analysis methods such as classification, target detection and anomaly detection have emerged. Among them, hyperspectral anomaly detection (HAD), which performs unsupervised detection of objects that are spatially or spectrally different from the surrounding background, has received increasing attention in practical applications [12–16]. However, due to some real-world resource constraints such as lack of prior knowledge, low spatial resolution [17], lack of labels and insufficient sample sets bring many difficulties to HAD.

Existing methods are mainly divided into two categories: traditional methods and deep neural network (DNN)-based methods. Assuming that the background obeys multivariate Gaussian normal distribution, the Reed Xiaoli (RX) algorithm [18], by Reed and

Yu, has been proposed. If an RX detector estimates the background model using local statistics, this is an improved variant called local RX (LRX) [19]. Other traditional algorithms such as CRD [20], SRD [21], LSMAD [22] and LRASR [23] construct models by characterizing local neighbor pixels. These traditional methods have limited ability to characterize high-dimensional and complex data, thus making it difficult to achieve performance improvements. Under this circumstance, several efforts dedicated to determining hyperspectral anomalies by DNNs without any prior knowledge have been developed. DNN-based methods relying on AE [17] and GAN [24] focus on minimizing the error of each spectral vector in the original image and reconstructed image. Other DNN-based methods like CNN [25] take local similarity of spectra into consideration. Typically, these methods either interest a single spectral vector or consider local spectral similarity. However, they usually suffer from ignoring the topological relationship in HSIs. For instance, as shown in Figure 1, some anomalies are spectrally similar, whereas they are far apart in spatial dimensions, which makes most algorithms falsely consider anomalies as background. In addition, the blind introduction of hand-craft topological relationships may also degrade the model capability and generalizability in handling diverse HSIs.

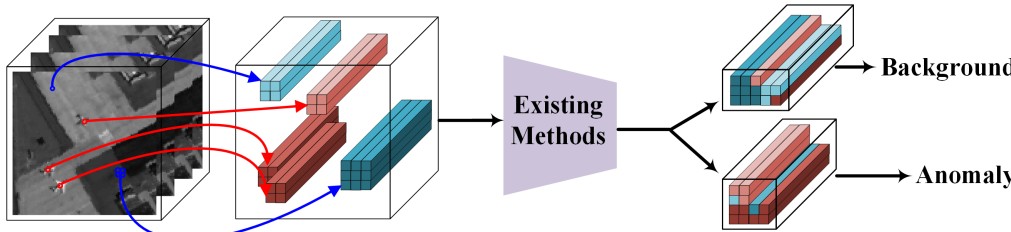

**Figure 1.** An illustration of the existing methods. Some anomalies (red blocks) are far apart in spatial dimensions and, thus, may be considered as background (blue blocks).

To address the aforementioned limitations, based on the observation that some anomalies are very far apart in the spatial domain, we put forward a relationship attention network for HAD, named RANet. Instead of hand-craft prior, RANet adopts an attention module that learns deep topological relationships from original HSIs in an end-to-end manner, where the attention intensity is self-adaptively controlled by adjacency reconstruction ability, which can effectively reduce human intervention. In this way, different categories have different topological relationships, which can better adapt to the characteristics of HSIs to achieve end-to-end personalized model learning. Furthermore, to address the difficulty in defining the neighbor relationship of an anomaly in a high-dimensional space, we leverage a graph attention network (GAT) with a customized incidence matrix to drive the non-local topological relationship between anomalies more significant without the need to know the structure of the whole HSIs. In particular, the customized incidence matrix, which utilizes spectral angle distance (SAD) instead of Euclidean distance (ED) to calculate the distance between spectral vectors, is imposed to better capture spectral similarity. More importantly, to make full use of correlations among adjacency, we establish an unsupervised CAE to determine anomalies, which is jointly learned with the topological relationship attention to achieve promising model accuracy.

The main contributions of our work are as follows:

- We propose a novel framework, RANet, for hyperspectral anomaly detection. To the best of our knowledge, this is the first attempt to explore the potential of topological relationships in this task;
- We introduce a customized incidence matrix that directs GAT to pay attention to topological relationships in HSIs, where the attention intensity is self-adaptively adjusted to different data characteristics;
- Furthermore, an end-to-end unsupervised CAE with high-fidelity and high-dimensional data representation is developed as the reconstruction backbone;

- We jointly learn the reconstructed backbone and topological attention to detect anomalies with the reconstruction error. Extensive experiments on HSIs indicate that our RANet outperforms existing state-of-the-art methods.

## 2. Related Work

Deep neural networks (DNNs) have attracted much attention in the field of HAD because they can estimate complex functions through hierarchical structures and capture hidden features; thus, DNN-based anomaly detection technologies have been applied to many fields [26–29]. Among them, unsupervised learning models have become mainstream research topics because they do not require label information. Autoencoders (AEs) can achieve automatic selection of features and improve the interpretability of networks. To tackle the problems of high dimensionality, redundant information and degenerate bands of hyperspectral images, a spectral constraint AAE (SC_AAE) [30] incorporates a spectral constraint strategy into adversarial autoencoders (AAE) to perform HAD. It introduces the spectral angular distance in the loss function to enhance spectral consistency. Liu et al. [31] introduced a dual-frequency autoencoder (DFAE) detection model to enhance the separability of background anomalies while breaking the dilemma of limited generalization ability in no-sample HAD tasks. An unsupervised low-rank embedded network (LREN) [6] estimates residuals efficiently in deep latent spaces by searching the lowest-rank representation based on a representative and discriminative dictionary. Additionally, Xie et al. [32] constructed a spectral distribution-aware estimation network (SDEN) that does not conduct feature extraction and anomaly detection in two separate steps but instead learns both jointly to estimate anomalies. However, since AEs perform feature learning for each spectral vector, the similarity between local spectral vectors is not considered. Convolutional autoencoders (CAEs) pay attention to local similarity by combining the convolution and pooling operations of the convolutional neural network and realize a deep neural network through stacking. Considering that the structure of the convolutional image generator for hyperspectral images can capture a large number of image statistics, Auto-AD [33] designs a network to reconstruct the background through a fully convolutional AE skip connection. The main advantage of generative adversarial network (GAN) is that it surpasses the functions of traditional neural network classification and feature extraction and can generate new data according to the characteristics of real data. Jiang et al. [34] introduced a weakly supervised discriminant learning algorithm based on spectrally constrained GAN, which utilizes background homogenization and anomaly saliency to enhance the ability to identify anomalies and backgrounds when the anomaly samples are limited and sensitive to the background. Based on the assumption that the number of normal samples is much larger than the number of abnormal samples, HADGAN [35] proposed a generative adversarial network for HAD under unsupervised discriminative reconstruction constraints. Fu et al. [36] proposed a new solution using the plug-and-play framework. To be more specific, by implementing a plug-in framework, the denoiser is employed as a prior for the representation coefficients, while a refined dictionary construction method is suggested to acquire a more refined background dictionary. Introducing BSDM (Background Suppression Diffusion Model), Ma et al. [37] proposed a novel solution for HAD that enables the simultaneous learning of latent background distributions and generalization to diverse datasets, facilitating the suppression of complex backgrounds. Wang et al. [38] proposed a tensor low-rank and sparse representation method for HAD. They put forward a strategy for constructing dictionaries that relies on the weighted tensor kernel norm and $L_{F,1}$ sparse regularization norm, aiming to separate low-rank backgrounds from outliers. Although the above-mentioned methods seem to achieve good performance, ignoring topological relationships limits the improvement of detection accuracy.

## 3. Proposed Method

In this section, we propose RANet for HAD. We describe the overall structure of RANet in Section 3.1, followed by the topological-aware module in Section 3.2. Then, the details of reconstructed backbone are given in Section 3.3 and, finally, the joint learning part in Section 3.4.

### 3.1. Overall Architecture

An HSI dataset $\mathbf{Y} \in \mathbb{R}^{Z \times M \times N}$ with $Z$ spectral bands and $M \times N$ pixels in spatial domain is denoted as $\mathbf{Y} = [y_1, y_2, ..., y_{M \times N}]$, where $y_i \in \mathbb{R}^{Z \times 1}$ represents the spectral vector of the $i$th pixel. $\mathbf{Y}$ consists of a background sample set $\mathbf{Y}_B$ and an anomaly sample set $\mathbf{Y}_A$, which have different characteristics in both spectral and spatial domains, i.e., $\{\mathbf{Y} = \mathbf{Y}_B \cup \mathbf{Y}_A\} \wedge \{\mathbf{Y}_B \cap \mathbf{Y}_A = \varnothing\}$. Since DNN-based models are data-driven and $\mathbf{Y}_B$ has much more learnable samples than $\mathbf{Y}_A$, existing DNN-based unsupervised HAD methods usually focus on learning the representation of each sample $y_i \in \mathbb{R}^{Z \times 1}$, which leads to small reconstruction errors in background samples and large reconstruction errors in anomaly samples. This process essentially constructs a model $M(\cdot)$ that indicates the quality of reconstruction, which can be expressed as

$$\hat{\mathbf{Y}} = M(y_i \in \mathbf{Y}; \theta) \tag{1}$$

where $\hat{\mathbf{Y}}$ is the reconstructed HSI and $\theta$ represents the parameters of the reconstruction model $M(\cdot)$, which are learned from the intrinsic characteristics of each sample $y_i$.

However, these methods may not be ideal because they only pay attention to the latent features of each sample and ignore the connections between samples, especially those that are spatially distant. With these in mind, we introduce the topological relationship into the classical reconstruction model $M(\cdot)$, which serves as an attention mechanism to make the model achieve better representation learning. Formally, we define our reconstruction model as

$$\hat{\mathbf{Y}} = M\big(\Phi(y_i, y_j), y_{i,j} \in \mathbf{Y}; \theta\big) \tag{2}$$

where $\theta$ is learned not only from each sample $y_i$, but also from the topological relationships $\Phi(y_i, y_j)$ that exist between different samples.

Considering that it is a time-consuming task to calculate the topological relationship between each sample and that the spectral vectors within the local region of HSI have high similarity, we divide an HSI $\mathbf{Y} \in \mathbb{R}^{Z \times M \times N}$ with $M \times N$ into $P$ small 3D cubes and denote them as $\mathbf{H} = \{h_i\}_{i=1}^{P}$. Here, $h_i$ represents the $i$th cube, consisting of $m \times n$ spectral vectors, i.e., $h_i = [y_1, y_2, \ldots, y_{m \times n}], (m \ll M, n \ll N)$. Therefore, our model is modified as

$$\hat{\mathbf{H}} = M\Big(\Phi(h_i, h_j), y_i \in \{h_i\}_{i=1}^{p}; \theta\Big). \tag{3}$$

The learning objective of our model $M(\cdot)$ is that the reconstructed background samples outperform the abnormal samples, such that the difference between the original $\mathbf{H}$ and the reconstructed $\hat{\mathbf{H}}$ represents anomalies to be detected. Figure 2 shows a high-level overview of the proposed RANet, which is composed of three major components: topological-aware module, reconstructed backbone and joint learning. Next, we introduce how to design and learn our reconstruction model $M(\cdot)$ with these three components.

### 3.2. Topological-Aware Module

The topological-aware (TA) module intends to dig out deep topological relationships embedded in the customized incidence matrix to guide the reconstruction backbone for high-fidelity background sample representation. In other words, the TA module attempts to acquire the topological relationships of HSI, i.e., $\Phi(h_i, h_j)$, and inject them as attention intensity into the reconstruction model $M(\cdot)$. Given that HAD lacks prior knowledge, we let the TA module learn in an unsupervised manner, which can be subdivided into

three parts: customized incidence matrix, topological feature extraction and decoding for reconstruction.

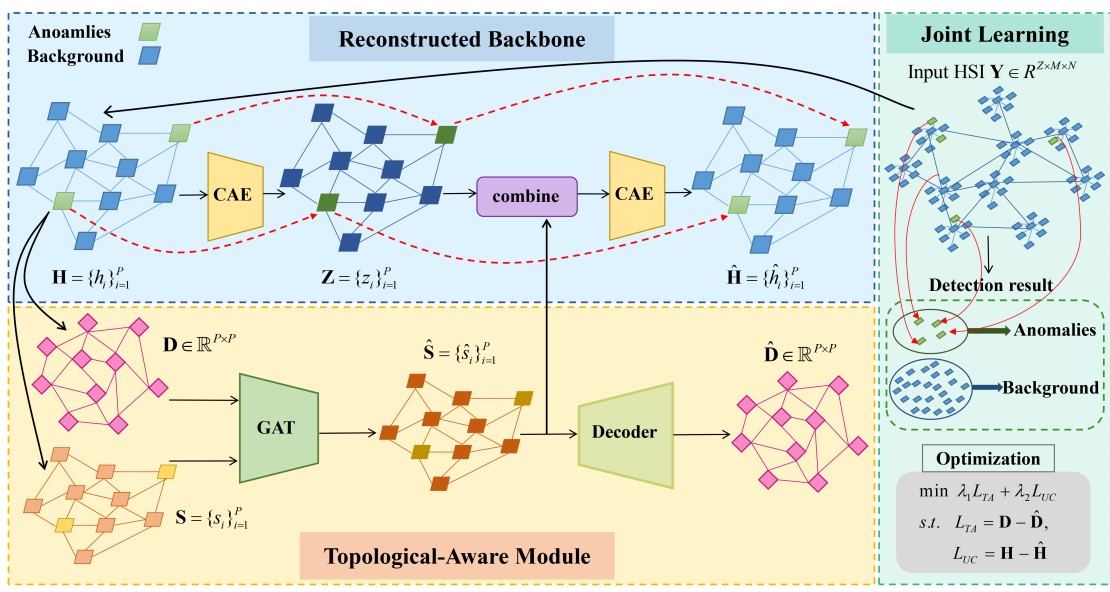

**Figure 2.** High-level overview of the proposed RANet. For the topological-aware (TA) module part, we first use an incidence matrix D and a representation set S as input to GAT, to obtain a topological feature map $\hat{\mathbf{S}}$. The reconstruction of D (denoted as $\hat{\mathbf{D}}$) is gained after the decoding process. As for the reconstructed backbone part, we then use CAE as our network backbone, combined with the previously obtained $\hat{\mathbf{S}}$, to reconstruct H into $\hat{\mathbf{H}}$. The joint learning part is responsible for jointly learning the TA module and the reconstructed backbone in an end-to-end manner.

The customized incidence matrix is used to reflect neighbor relationship of HSI's cube set $\mathbf{H} = \{h_i\}_{i=1}^P$. Concretely, taking the inherent spectral characteristics of HSI into consideration, we employ the spectral angle distance (SAD) instead of Euclidean distance as the similarity measure to determine the neighbor cubes of each cube. The higher the similarity between two cubes, the more likely they are to be neighbors. The SAD depicts the included angle of two spectral vectors. A smaller SAD value indicates that the spectral vectors are more similar. As shown in Figure 3, two spectral curves from the homogeneous background may have large Euclidean distances even if their shapes are highly similar. On the contrary, their SAD is not affected by the difference in absolute gray values. Therefore, it is reasonable to use SAD to determine the neighbor relationship of $\mathbf{H}$ in the HAD task. Formally, the incidence matrix can be described as $\mathbf{D} = \{d_{ij}\}_{i=1,j=1}^{i=P,j=P}$, where $d_{ij}$ represents the similarity between the $i$th and $j$th cubes. Thus, the construction of $\mathbf{D}$ is formulated as

$$\mathbf{D} = \begin{cases} 0, d_{i,j} \geq \eta \\ 1, d_{i,j} < \eta \end{cases} , \text{where}$$
$$d_{i,j} = \text{SAD}(h_i, h_j) = \arccos \frac{h_i^{\mathrm{T}} h_j}{\|h_i\|_2 \|h_j\|_2}. \tag{4}$$

Here, $h_i$ and $h_j$ are reshaped as $Z \times m \times n$-dimension vectors before calculation. $\eta$ represents the threshold to determine how close two cubes are to be considered neighbors; its value is discussed in experiments part.

The topological feature extraction is achieved by a graph attention network (GAT). For easier and more efficient operation, we represent each cube $h_i$ by the average of all spectral vectors in that cube according to local similarity and generate a representation set

$\mathbf{S} = \{s_i\}_{i=1}^{P}$, where $s_i \in \mathbb{R}^{Z \times 1}$. Thus, the topological features of the entire HSI are obtained by the weighted sum of the representation of each cube and its neighbors

$$\hat{\mathbf{S}} = \{\hat{s}_i\}_{i=1}^{P} = \left\{ \sum_{k \in N_i} \varphi_{i,k} s_k \right\}_{i=1}^{P} \tag{5}$$

where $\hat{\mathbf{S}}$ represents the topological feature, which is combined with the reconstructed backbone later. $N_i$ indicates the neighbor set of the cube representation $s_i$, which can be retrieved by the customized incidence matrix $\mathbf{D}$. $s_k$ means the neighbor of $s_i$, and $\varphi_{i,k}$ is the weight coefficient of this neighbor.

Since different neighbors have different importance, the weight coefficient of each neighbor is acquired using the GAT learning:

$$\varphi_{i,k} = GAT(s_i, s_k). \tag{6}$$

Specifically, we first apply a shared linear transformation matrix $W_{Gat}$ to standardize the representation vector of each cube. Then, a single-layer neural network parameterized by $a_{Gat}$ and $W_{Gat}$ is established with the nonlinear activation function LeakyReLU:

$$e_{i,k} = \text{LeakyReLU}\left[a_{Gat}^{T}(W_{Gat}s_i \| W_{Gat}s_k)\right] \tag{7}$$

where $\|$ means the concatenation operation and $e_{i,k}$ represents the weight coefficient of $s_k$ to $s_i$. To normalize the weight coefficient among different cubes, a Softmax function is employed to obtain the final weight coefficient $\varphi_{i,k}$:

$$\begin{aligned}
\varphi_{i,k} = \text{Softmax}(e_{i,k}) &= \frac{\exp(e_{i,k})}{\sum_{k \in N_i} \exp(e_{i,k})} \\
&= \frac{\exp\left(\text{LeakyReLU}\left[a_{Gat}^{T}(W_{Gat}s_i \| W_{Gat}s_k)\right]\right)}{\sum_{k \in N_i} \exp\left(\text{LeakyReLU}\left[a_{Gat}^{T}(W_{Gat}s_i \| W_{Gat}s_k)\right]\right)}.
\end{aligned} \tag{8}$$

The decoding process is designed to learn the parameters $a_{Gat}$ and $W_{Gat}$, which is inspired by the autoencoder network. Different from the general autoencoder network, we design a simple and effective decoding method for reconstruction based on data features of the incidence matrix $\mathbf{D}$, including dimensional and numerical, without introducing additional learnable parameters. The decoding process is defined as

$$\hat{\mathbf{D}} = Sigmoid\left(\hat{\mathbf{S}}^{T} \times \hat{\mathbf{S}}\right) \tag{9}$$

where $\times$ denotes matrix multiplication and $\hat{\mathbf{D}} \in \mathbb{R}^{P \times P}$ refers to the reconstruction of $\mathbf{D}$. Consequently, after optimizing the learnable parameters $a_{Gat}$ and $W_{Gat}$ by minimizing the reconstruction error $\mathbf{D} - \hat{\mathbf{D}}$, the topological feature $\hat{\mathbf{S}}$ is ready for the reconstruction backbone.

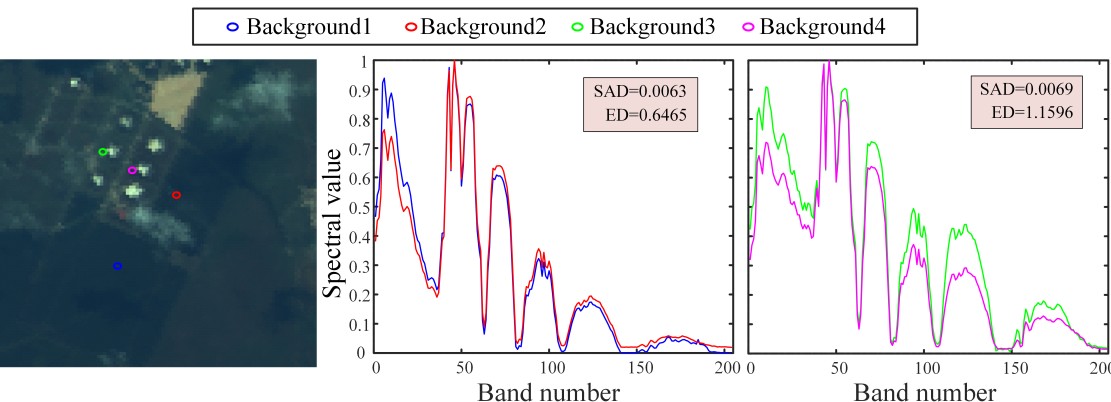

**Figure 3.** (**Left**): Locations of the background samples in the pseudo-color image. (**Middle**) and (**Right**): Spectral curves of the background samples in the corresponding color. The legend indicates the spectral angle distance (SAD) and the Euclidean distance (ED) of the two spectral curves.

### 3.3. Reconstructed Backbone

The backbone of our reconstruction model $M(\cdot)$ is configured with an autoencoder. Compared with a fully connected autoencoder, a convolutional autoencoder (CAE) can extract latent features of HSI under the condition of local perception and parameter sharing. This means that CAE can capture the connection among local data to generate semantic features, thereby improving detection performance. Hence, we employ a multi-layer stacked combination of convolutional layers and nonlinear activation functions as our reconstructed backbone.

The reconstructed backbone consists of an encoder $En(\cdot)$ and a decoder $De(\cdot)$. $En(\cdot)$ is employed to learn the hidden representation of HSI cube set $\mathbf{H} = \{h_i\}_{i=1}^{P}$ and generate the hidden feature set $\mathbf{Z} = \{z_i\}_{i=1}^{P}$:

$$
\begin{aligned}
\mathbf{Z} &= En(\mathbf{H}) \\
&= f_{En}(W_{En} * \mathbf{H} + b_{En})
\end{aligned}
\tag{10}
$$

where $f_{En}$ signifies the ReLU function and $*$ denotes convolution operation with learnable weight matrix $W_{En}$ and bias vector $b_{En}$. Instead of feeding $\mathbf{Z}$ directly into the decoder $De(\cdot)$, we combine this latent feature of HSI with the topological features $\hat{\mathbf{S}}$ together to guide $De(\cdot)$ for reconstruction. Thus, the decoding process in our reconstructed backbone can be formulated as

$$
\begin{aligned}
\hat{\mathbf{H}} &= De(\hat{\mathbf{S}} \circ \mathbf{Z}) \\
&= f_{De}(W_{De} * (\hat{\mathbf{S}} \circ \mathbf{Z}) + b_{De})
\end{aligned}
\tag{11}
$$

where $\circ$ is the Hadamard product after broadcasting on $\hat{\mathbf{S}}$ and $\hat{\mathbf{H}}$ is the reconstructed HSI mapped by the ReLU function $f_{De}$ and the convolution kernel with learnable parameters $W_{De}$ and $b_{De}$.

### 3.4. Joint Learning

To avoid falling into a suboptimal solution caused by the separation of feature extraction and anomaly detection [39], the TA module and the reconstructed backbone jointly learn in an end-to-end manner. With the gradient-descent-based joint optimization, the learning process of topological relations and the data reconstruction process can constitute a unified framework, as expected from our established model (Equation (3)). Therefore, the objective function of the proposed model $M(\cdot)$ is defined as

$$
\begin{aligned}
\min \quad & \lambda_1 L_{TA} + \lambda_2 L_{UC} \\
s.t. \quad & L_{TA} = \mathbf{D} - \hat{\mathbf{D}}, L_{UC} = \mathbf{H} - \hat{\mathbf{H}}.
\end{aligned}
\tag{12}
$$

Here, $L_{TA}$ demonstrates the loss function of the TA module referring to the reconstruction error of the customized incidence matrix and $L_{UC}$ represents the mean squared error (MSE) of the original HSI dataset, which is employed to optimize the reconstructed backbone. $\lambda_1$ and $\lambda_2$ control the proportion of corresponding terms in the objective function and the values are discussed in the experiments part.

As discussed in our model (Equation (3)), the parameters $\theta = \{W_{Gat}, a_{Gat}, W_{En}, b_{En}, W_{De}, b_{De}\}$ of the model $M(\cdot)$ are learned by minimizing the objective function, which allows our RANet to have better reconstruction ability for background samples to detect anomalies as

$$
\begin{aligned}
\mathbf{Y}_A &= \mathbf{H} - \hat{\mathbf{H}} \\
&= H - M\big(\Phi(h_i, h_j), y_i \in H; \theta\big)
\end{aligned} \tag{13}
$$

## 4. Experimental Results

### 4.1. Experimental Setup

**Dataset Description.** We evaluate our RANet on four benchmark hyperspectral datasets, including Texas Coast-1, Texas Coast-2, Los Angeles, and San Diego, with noisy bands being removed and reference maps of the original data being manually labeled. Texas Coast-1 dataset was recorded from the Texas Coast, USA, in 2010 by the AVIRIS sensor. The image scene covers an area of $100 \times 100$ pixels, with 204 spectral bands and a 17.2 m spatial resolution. Texas Coast-2 dataset was obtained in the same location as Texas Coast-1, except that there are 207 spectral bands. Los Angeles dataset was acquired by the AVIRIS sensor over the area of Los Angeles city. After removing the noisy bands, this dataset, including 205 spectral channels, has a spatial size of $100 \times 100$ with a ground resolution of 7.1 m. There are some houses considered as anomalies in these three datasets. The San Diego dataset contains widely used hyperspectral images, which were collected by the AVIRIS sensor over the San Diego airport area, CA, USA. This image contains $100 \times 100$ pixels, with 189 spectral bands in wavelengths ranging from 400 to 2500 nm. We consider three airplanes as the anomalies to be detected in this dataset.

**Evaluation Criterion.** We employ the receiver operating characteristic (ROC) [40] curve, the area under ROC curve (AUC) [41] and Box–Whisker Plots [42] as our evaluation metrics to quantitatively assess the anomaly detection performance of RANet and its comparison algorithms. ROC curve can be plotted by the true positive rate (TPR) and the false positive rate (FPR) at various thresholds $\tau$ based on the ground truth. AUC acts as an evaluation metric to measure the performance of the detector by calculating the whole area under the ROC curve. The closer the AUC of (TPR, FPR) value is to 1, the better the detection performance. Finally, the Box–Whisker Plots are used to indicate the degree of background suppression and separation from the anomaly.

**Implementation Details.** In our RANet, we focus on four parameters: the threshold $\eta$, the number of hidden nodes and the hyperparameters $\lambda_1$ and $\lambda_2$. As a pivotal parameter, the value of threshold $\eta$ greatly impacts the determination of the customized incidence matrix. With a small value, the significant topological information provided by the customized incidence matrix will be very limited, since only a few cubes are considered neighbors. However, with a large value, RANet may mistake the background for an anomaly. Hence, we set the number of thresholds $\eta$ to 0.001, 0.01, 0.05, 0.1, 0.2 and 0.5. As shown in Figure 4a, all the datasets obtain the best detection results when the number of threshold $\eta$ reaches 0.05. As for the number of hidden nodes, it plays a crucial role in the reconstructed backbone. With an appropriate value, the reconstructed backbone can effectively extract features embedded in the original input space. Therefore, we set the value to 8, 10, 12, 14, 16 and 18 and evaluate the AUC scores of (TPR, FPR) on each HSI. As shown in Figure 4b, all the datasets achieve the best performance when the number of hidden nodes is set to 10. In the objective function, there are two hyperparameters (i.e., $\lambda_1$ and $\lambda_2$) corresponding to two modules. With an inappropriate value of $\lambda_1$, the TA model can maximize the advantage of adjacency reconstruction ability. In addition, the setting of $\lambda_2$ plays a significant role in the reconstructed backbone and a proper value can guarantee

the reconstruction capability of the reconstructed backbone on the original HSI. We set the hyperparameter $\lambda_1$ to 0.5, 0.6, 0.7, 0.8 and 0.9 and the hyperparameter $\lambda_2$ to 0.1, 0.2, 0.3, 0.4 and 0.5. According to the 3D diagrams plotted in Figure 5, the optimal values are selected as 0.7 and 0.3. We train RANet with the SGD optimizer on NVIDIA GeForce RTX 3090 Ti in an end-to-end manner, setting the learning rate to $10^{-3}$ and the epoch to 2000. Each set of experiments is performed 10 times, the best results taken.

### 4.2. Detection Performance

We compare RANet with seven frequently cited and state-of-the-art approaches, including RX [18], LRASR [23], LSMAD [22], SSDF [43], LSDM–MoG [44], PTA [45], PAB–DC [46], Auto-AD [33] and 2S–GLRT [47]. Reed Xiaoli (RX) algorithm is proposed assuming that the background obeys multi-variate Gaussian normal distribution. LRASR is a hyperspectral anomaly detection method based on the existence of mixed pixels, which assumes that background data is located in multiple low-rank subspaces. LSMAD realizes anomaly detection by decomposing hyperspectral data into low-rank background components and sparse anomaly components. SSDF is a forest discriminant method based on subspace selection for hyperspectral anomaly detection. LSDM–MoG is a low-rank sparse decomposition method based on a mixed Gaussian model for HAD. PTA proposed a tensor approximation method based on priors. PAB–DC is an HAD method based on low rank and sparse representation strategy. Auto-AD is a kind of method based on an autonomous hyperspectral anomaly detection network. 2S–GLRT proposes an adaptive detector based on generalized likelihood ratio test (GLRT). All of the methods above are reimplemented according to their papers and open-source codes.

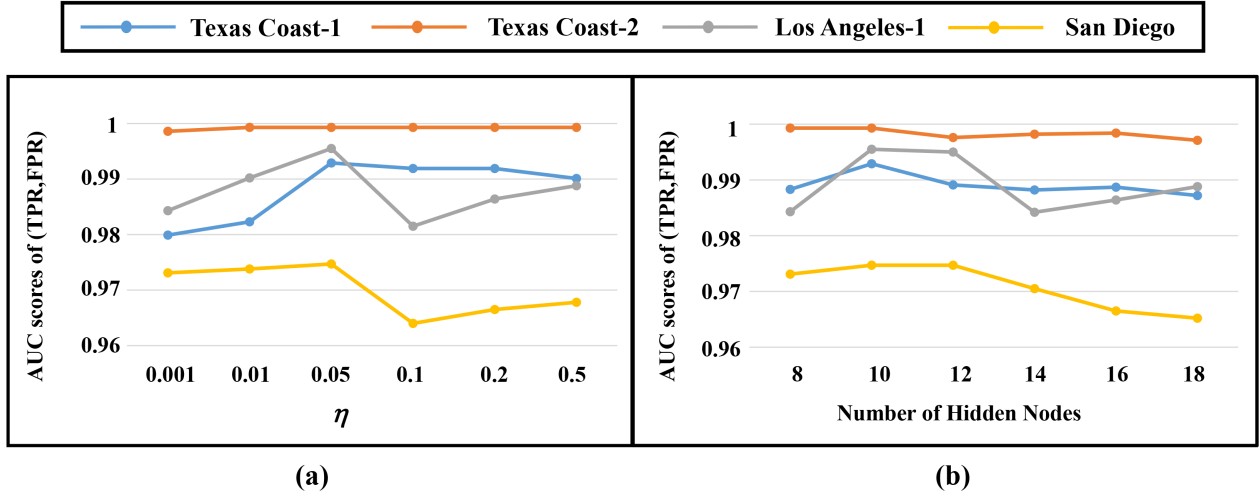

**Figure 4.** Implementation details on four HSIs. (**a**) The threshold $\eta$. (**b**) Number of hidden nodes.

In order to visually demonstrate the detection performance, Figure 6 displays the visual detection maps of the four HSIs under the above-mentioned methods. It is worth noting that RANet can completely and accurately detect anomalies of different sizes and locations, while other methods have problems such as false detection, missed detection and blurred targets to varying degrees. In addition, RANet can effectively suppress strip noise, thereby achieving good performance.

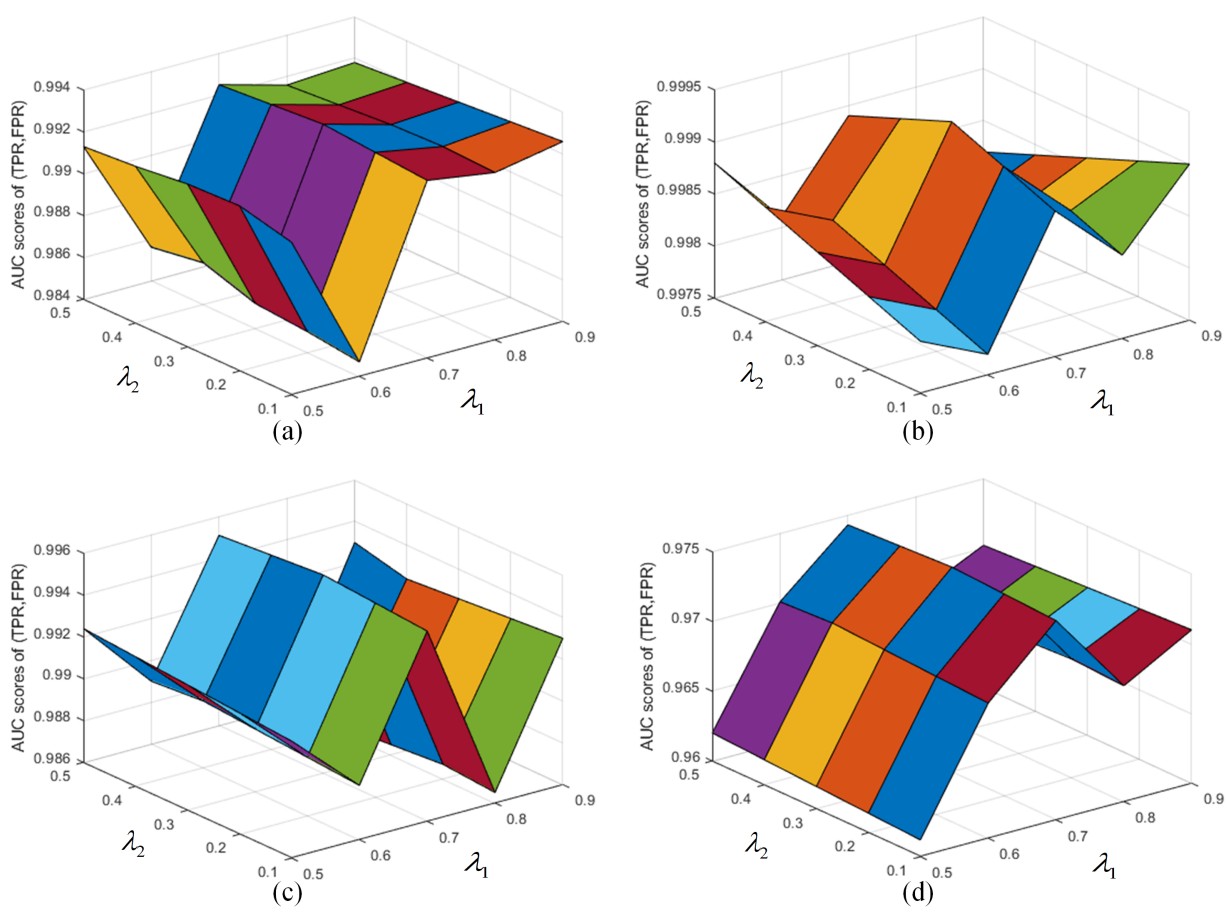

**Figure 5.** Implementation details of the hyperparameters $\lambda_1$ and $\lambda_2$ on (**a**) Texas Coast-1, (**b**) Texas Coast-2, (**c**) Los Angeles and (**d**) San Diego. Different colors refer to different intervals of values of $\lambda_1$ and $\lambda_2$.

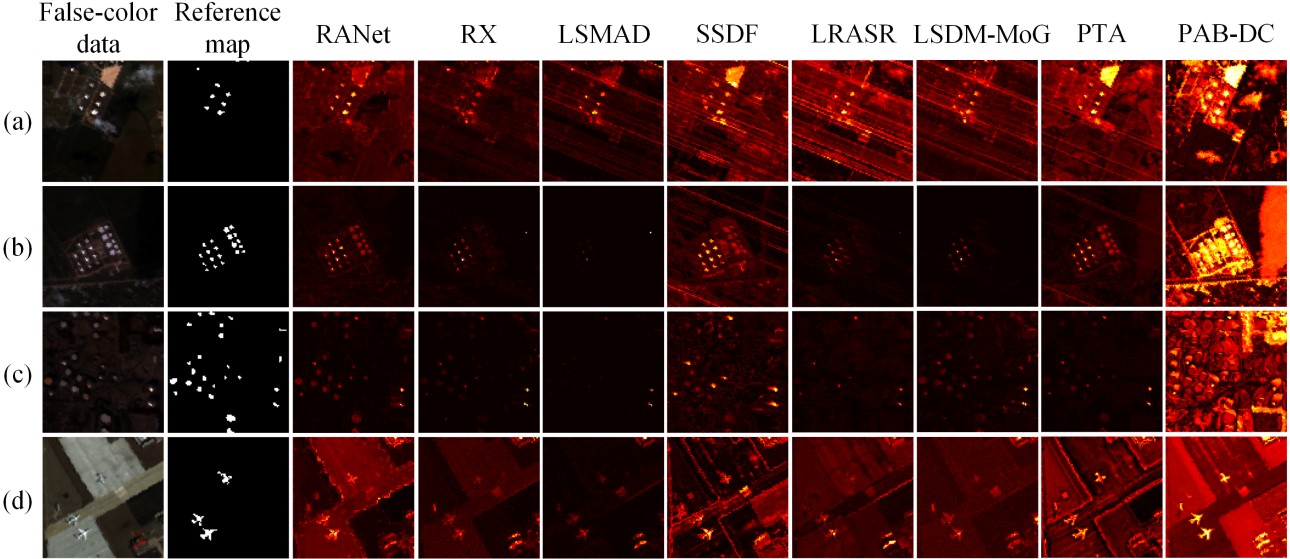

**Figure 6.** False-color data, reference map, and detection maps of the compared methods for (**a**) Texas Coast-1, (**b**) Texas Coast-2, (**c**) Los Angeles and (**d**) San Diego.

By quantitatively comparing these detectors, Table 1 lists the AUC scores of (TPR, FPR) of the seven popular methods and RANet on the four real HSIs. Notably, the AUC scores of (TPR, FPR) are consistent with the visual detection maps in Figure 6. It can be observed that RANet outperforms other methods with the highest AUC score of (TPR, FPR) on every HSI.

To further demonstrate the detection performance among compared methods and RANet, we plot the ROC curves of (TPR, FPR) of eight approaches on the four HSIs. As shown in Figure 7, the ROC curves of (TPR, FPR) of RANet lie nearer the top-left corner, and it is obvious that RANet can obtain a high probability of detection and provide the most excellent detection performance. As for RX, its curve is also very close to the upper left corner, but it still does not catch up with RANet's detection effect due to missing some objects. For other methods, they all show defects on different datasets, resulting in poor performance of the ROC curves.

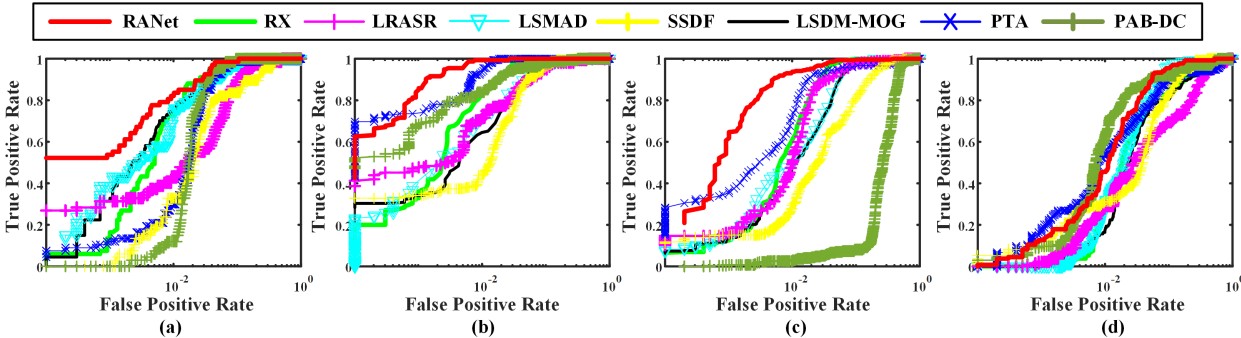

**Figure 7.** ROC curves of (TPR, FPR) for the algorithms on (**a**) Texas Coast-1, (**b**) Texas Coast-2, (**c**) Los Angeles and (**d**) San Diego.

Meanwhile, we employ the Box–Whisker Plots to analyze the ability to separate the anomalies and background and then observe the effect of suppressing the background. As illustrated in Figure 8, the detection results of each method correspond to two boxes, in which the red box represents the distribution range of anomaly detection values and the blue box represents the distribution range of background detection values. The relative positions and compactness of the boxes reflect the trends in background and anomalous pixel distributions. In general, RANet can evidently reveal the capability of discriminating anomaly and background.

**Table 1.** AUC scores of (TPR, FPR) for the compared methods on different datasets. The scores in bold form refer to the best performance.

|      | RANet | RX | LRASR | LSMAD | LSDM–MoG | PTA | SSDF | PAB–DC | Auto-AD | 2S–GLRT |
|------|-------|-----|-------|-------|----------|-----|------|--------|---------|---------|
| TC-1 | **0.9929** | 0.9907 | 0.9563 | 0.9829 | 0.991 | 0.9775 | 0.9466 | 0.9793 | 0.9906 | 0.9898 |
| TC-2 | **0.9993** | 0.9946 | 0.9798 | 0.9856 | 0.9845 | 0.998 | 0.9781 | 0.9912 | 0.9937 | 0.9913 |
| LA   | **0.9955** | 0.9887 | 0.9796 | 0.9804 | 0.9781 | 0.9738 | 0.9423 | 0.9323 | 0.9913 | 0.9915 |
| SD   | **0.9747** | 0.9403 | 0.8891 | 0.9689 | 0.931 | 0.9391 | 0.9454 | 0.9669 | 0.9530 | 0.9647 |
| Avg  | **0.9906** | 0.9786 | 0.9512 | 0.9795 | 0.9712 | 0.9721 | 0.9531 | 0.9674 | 0.9821 | 0.9843 |

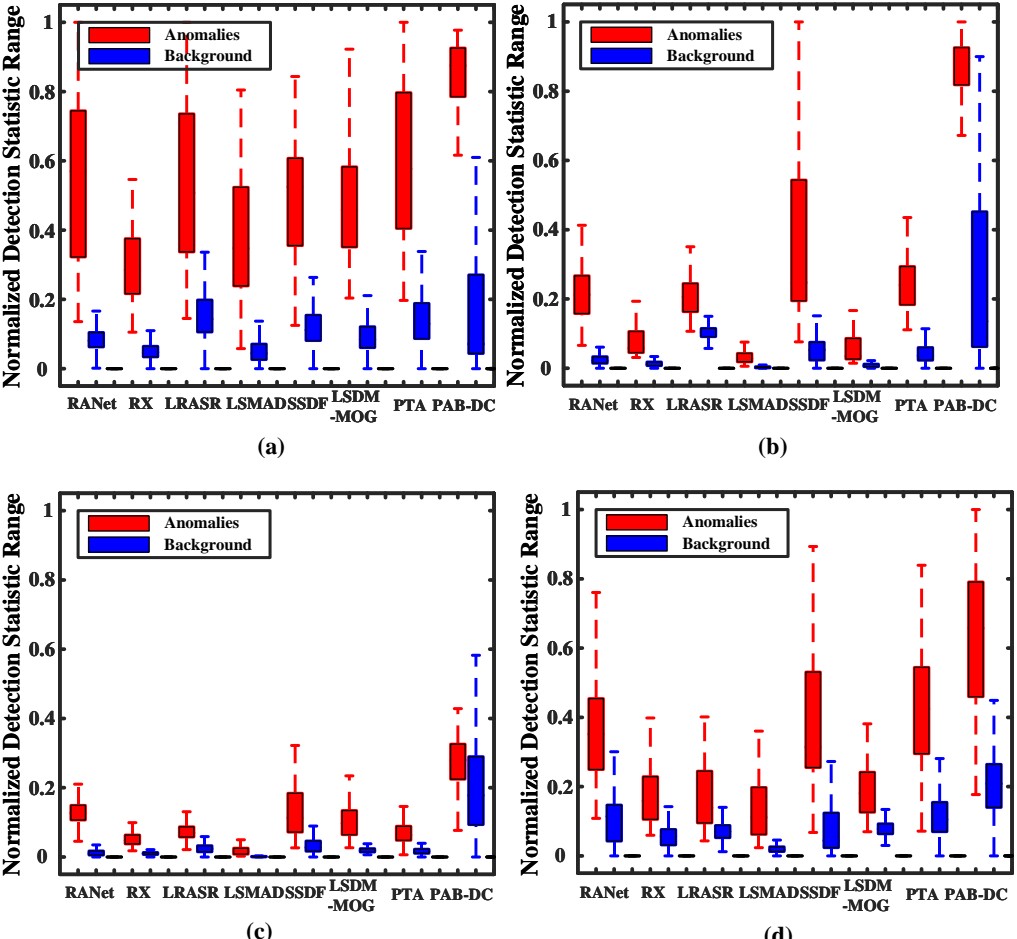

**Figure 8.** Background–anomaly separation analysis of the compared methods on (**a**) Texas Coast-1, (**b**) Texas Coast-2, (**c**) Los Angeles and (**d**) San Diego.

### 4.3. Discussion

In this section, we conduct experiments to discuss four factors related to the performance of RANet's framework.

**Network Architecture.** In order to explore the training performance of the reconstructed backbone, we adopt AE, CAE and both CAE and TA modules (CAE + TA) separately. As shown in Figure 9a, it is obvious that, with joint learning of CAE + TA, the framework achieves the most satisfying detection result. Since AE performs anomaly detection by extracting the hidden layer features of each spectral vector without considering the correspondence between similar spectra, it does not obtain high detection results. Instead, CAE takes advantage of local spectral similarity, which makes its detection performance better than AE. For all datasets, the detection results illustrate the effectiveness of the TA module.

**Determination of Set S.** As shown in Figure 9b, we compare the effectiveness of the classical clustering algorithm K-Means and spectral averaging for the set **S**. As the most classical clustering method, K-Means aims to select the most representative samples. In RANet, we perform clustering on each cube and select a representative vector for each cube. It is worth noting that spectral averaging is slightly better than K-Means clustering for the Texas Coast-1 dataset, Texas Coast-2 dataset and Los Angeles dataset. As for the San Diego dataset, spectral averaging shows superior performance than K-Means clustering, which indicates that spectral averaging can obtain a more suitable set **S** to make the performance of the TA module fully utilized.

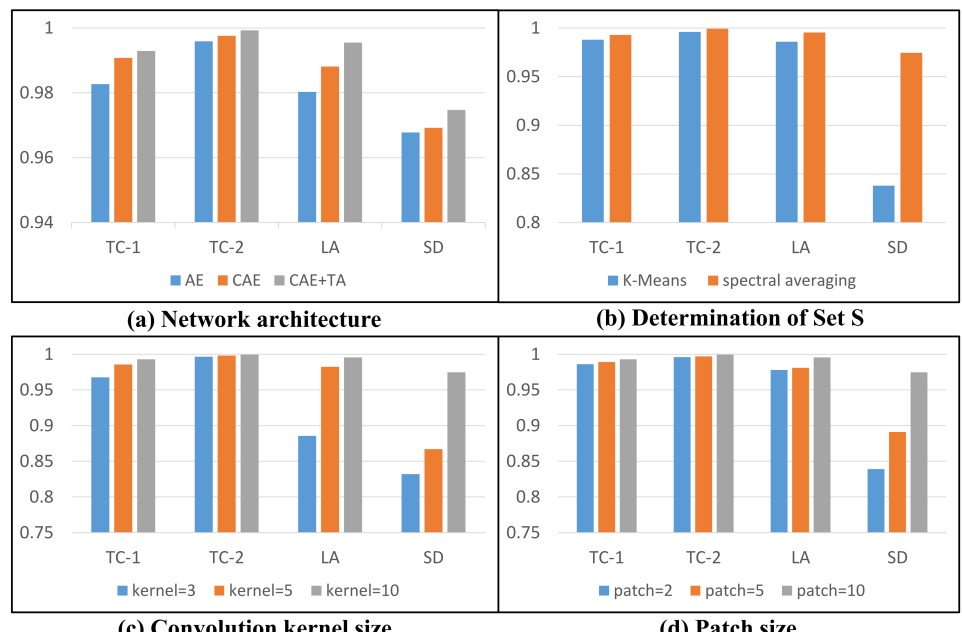

**Figure 9.** Detection accuracy comparison under four factors related to the performance of RANet on four HSIs.

**Convolution Kernel Size.** As one of the most important parameters in CAE, the size of the convolution kernel greatly affects the performance of the network. Therefore, we conduct an experiment to analyze the selection of the size of the convolution kernel. As shown in Figure 9c, we set the convolution kernel size to 3, 5 and 10 to analyze the impact on the detection results. It can be seen that the performance continues to improve with the increase in the convolution kernel and, when the size of the convolution kernel is 10, the performance is superior. Although the improvement in performance is small on the Texas Coast-1 dataset and Texas Coast-2 dataset, it is high on the Los Angeles dataset and San Diego dataset. Consequently, we adopt a convolution kernel size of 10.

**Size of 3D Cube.** As defined in Section 3, the size of each cube determines the number of cubes, which in turn determines the detection performance of RANet. Hence, it is necessary to conduct an experiment to determine the size of a 3D cube. As shown in Figure 9d, we measure the detection performance when the cube size is set to 2, 5 and 10. The experiments show that the performance tends to be more competitive as the cube size increases and all the datasets achieve satisfactory performance with a cube size 10. Since the four HSIs all contain $100 \times 100$ pixels, the number of cubes $P$ is 100.

## 5. Conclusions and Future Work

In this paper, we propose a novel framework named RANet for HAD. The main intent of our method is to achieve relationship attention-guided unsupervised learning with CAE. First of all, RANet leverages an attention-based architecture with a customized incidence matrix to learn deep topological relationships from HSIs. Second, an unsupervised CAE is designed as the reconstructed backbone with high-fidelity high-dimensional data representations. Third, the reconstructed backbone and topological attention are jointly learned to obtain reconstructed hyperspectral images. Finally, the reconstruction error is used to detect anomalies. Extensive experiments are conducted to validate that our RANet has competitive performance when compared to state-of-the-art approaches. In the future, we plan to work on improving the way we build our models to reconstruct equally good models with less computation or fewer resources.

**Author Contributions:** Conceptualization, Y.S. and Y.L. (Yang Liu); methodology, Y.S. and X.G.; software, Y.S. and L.L.; validation, Y.W. and Y.Y.; writing—original draft preparation, Y.S. and Y.L. (Yunsong Li); writing—review and editing, Y.D. and M.Z.; supervision, Y.L. (Yunsong Li). All authors have read and agreed to the published version of the manuscript.

**Funding:** This work was supported by the National Natural Science Foundation of China under Grant 62121001.

**Data Availability Statement:** No new data were created.

**Conflicts of Interest:** The authors declare no conflict of interest.

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
