# Peer review of "RANet: Relationship Attention for Hyperspectral Anomaly Detection"

_remotesensing, doi:10.3390/rs15235570_

Round 1

Reviewer 1 Report

Comments and Suggestions for Authors

This manuscript proposes a relationship attention guided unsupervised learning with convolutional autoencoders (CAEs) for HAD, called RANet. Anomaly detection from hyperspectral images is an interesting and challenging problem for the committee. Generally, this paper is technically correct. However, I have some major concerns about the methodology and experiments that must be explained and solved. The major concerns for the manuscript are given in detail as follows.

1. In the related work, some related new matrix factorization-based methods and deep learning-based methods are missing. For example,

[1] M. Wang, Q. Wang, D. Hong, S. K. Roy and J. Chanussot, "Learning Tensor Low-Rank Representation for Hyperspectral Anomaly Detection," in IEEE Transactions on Cybernetics, vol. 53, no. 1, pp. 679-691, Jan. 2023.

[2]  X. Shen, H. Liu, J. Nie and X. Zhou, "Matrix Factorization With Framelet and Saliency Priors for Hyperspectral Anomaly Detection," in IEEE Transactions on Geoscience and Remote Sensing, vol. 61, pp. 1-13, 2023.

[3] X. Fu, S. Jia, L. Zhuang, M. Xu, J. Zhou and Q. Li, "Hyperspectral Anomaly Detection via Deep Plug-and-Play Denoising CNN Regularization," in IEEE Transactions on Geoscience and Remote Sensing, vol. 59, no. 11, pp. 9553-9568, Nov. 2021.

2. The motivation for introducing topological relationships to improve the performance of anomaly detection is still not explained clearly.

3. The authors compare the effectiveness of the classical clustering algorithm K-Means and spectral averaging for the set S in Figure 9 (b). However, the effectiveness of K-Means also depends on the preset number of clustering, what’s the number of clusters in the manuscript? Also, it seems not reasonable to compare only a preset number.

4. Also, the size of S should also be discussed.

5. Please provide the reference for PAB-DC.

6. The competitors are too old, it is suggested to add some state-of-the-art competitors to evaluate the proposed method, in particular some deep learning-based methods.

7. The parameters of the compared methods should be given.

8. Please include the computation time for all the methods to fully evaluate the proposed methods.

9. On line 350, the number of cubes P is 10?

10. Please have the manuscript thoroughly and carefully checked before submitting the manuscript. I have found several mistakes in this version.

Comments on the Quality of English Language

The quality of the English language can be further improved. Also, the manuscript should be carefully checked before submission.

Author Response

Responses for “RANet: Relationship Attention for Hyperspectral Anomaly

Detection”

Dear reviewer,

Thank you for reviewing our manuscript, and providing us with constructive feedback. We have taken your comments into consideration and made revisions to our manuscript. Additionally, we have provided a response to each comment, which is highlighted in blue in our submitted pdf file. Some of the large paragraph description changes are highlighted in yellow (some minor typos and description changes will not be noted) in our new submission. Here, the Summary of Changes is concluded as follows:

In the “Related Work” section, we agree that the description and introduction of some of the latest methods will help readers to have a more comprehensive and in-depth understanding of the HAD task, so we include some of the latest methods proposed in the past two years.

In the “Detection Performance” section, we have added two newer methods of comparison, namely Auto-AD and 2S-GLRT. In addition to this, we have added a paragraph to briefly describe each comparison method.

For Figure 8, we reached an agreement that a 2 by 2 format is better than a 1 by 4 one, since it is structurally more aesthetically pleasing and easier for the reader to read what is in it. So we changed Figure 8 to a 2 by 2 format to make it look clearer.

Thank you again for your time and attention. We hope that you will be satisfied with our revisions and responses.

Best regards,

The authors

Responses

This manuscript proposes a relationship attention guided unsupervised learning with convolutional autoencoders (CAEs) for HAD, called RANet. Anomaly detection from hyperspectral images is an interesting and challenging problem for the committee. Generally, this paper is technically correct.

Response:

Thanks for the reviewer’s positive comments. We sincerely appreciate that the reviewer agrees on the strengths of our paper, and thanks for your recognition of our research direction.

Comment (1)

In the related work, some related new matrix factorization-based methods and deep learning-based methods are missing.

For example,

[1] M. Wang, Q. Wang, D. Hong, S. K. Roy and J. Chanussot, "Learning Tensor Low-Rank Representation for Hyperspectral Anomaly Detection," in IEEE Transactions on Cybernetics, vol. 53, no. 1, pp. 679-691, Jan. 2023.

[2] X. Shen, H. Liu, J. Nie and X. Zhou, "Matrix Factorization With Framelet and Saliency Priors for Hyperspectral Anomaly Detection," in IEEE Transactions on Geoscience and Remote Sensing, vol. 61, pp. 1-13, 2023.

[3] X. Fu, S. Jia, L. Zhuang, M. Xu, J. Zhou and Q. Li, "Hyperspectral Anomaly Detection via Deep Plug-and-Play Denoising CNN Regularization," in IEEE Transactions on Geoscience and Remote Sensing, vol. 59, no. 11, pp. 9553-9568, Nov. 2021.

Response:

Thanks for the comment. In the revised manuscript, we have added some matrix factorization-based methods [1] and deep learning-based methods [2][3] in the related work.

[1] M. Wang, Q. Wang, D. Hong, S. K. Roy and J. Chanussot, "Learning Tensor Low-Rank Representation for Hyperspectral Anomaly Detection," in IEEE Transactions on Cybernetics, vol. 53, no. 1, pp. 679-691, Jan. 2023.

[2] X. Fu, S. Jia, L. Zhuang, M. Xu, J. Zhou and Q. Li, "Hyperspectral Anomaly Detection via Deep Plug-and-Play Denoising CNN Regularization," in IEEE Transactions on Geoscience and Remote Sensing, vol. 59, no. 11, pp. 9553-9568, Nov. 2021.

[3] J. Ma, W. Xie, Y. Li and L. Fang. BSDM: Background Suppression Diffusion Model for Hyperspectral Anomaly Detection[J]. arXiv preprint arXiv:2307.09861, 2023.

Comment (2)

The motivation for introducing topological relationships to improve the performance of anomaly detection is still not explained clearly.

Response:

Fig.1. Motivation for considering topological relationships.

Thanks for the valuable comment. The motivation for introducing topological relationships can be explained by the picture above. There are three aircraft in figure (a), A, B, and C, where B and C are closer together, and A is farther away from both of them. If the topological relationship is ignored, it can easily lead to the loss of the target as shown in Figure (b). Therefore, introducing topological relationships can lead to improving the performance of HAD tasks.

Comment (3)

The authors compare the effectiveness of the classical clustering algorithm K-Means and spectral averaging for the set S in Figure 9 (b). However, the effectiveness of K-Means also depends on the preset number of clustering, what’s the number of clusters in the manuscript? Also, it seems not reasonable to compare only a preset number.

Response:

We apologize for not articulating this part of the experiment clearly. We conducted a large number of experiments on the number of k-means clustering in advance, and selected the number of clustering with the best performance to compare with our spectral average method.

Comment (4)

Also, the size of S should also be discussed.

Response:

We apologize for not specifying the size of S. In our experiment, the size of S is 2. We have also added the description to our new submission.

Comment (5)

Please provide the reference for PAB-DC.

Response:

Thanks for your kind reminder. We are sorry we slipped up in the writing process. We have added the reference for PAB-DC to our new submission.

Comment (6)

The competitors are too old, it is suggested to add some state-of-the-art competitors to evaluate the proposed method, in particular some deep learning-based methods.

Response:

Thanks for the valuable comment. We added two new comparison methods in the experimental part, namely Auto-AD[4] and 2S-GLRT[5]. To say it in a word, Auto-AD is a kind of method based on an autonomous hyperspectral anomaly detection network. 2S-GLRT proposed an adaptive detector based on generalized likelihood ratio test (GLRT).

[4] S. Wang, X. Wang, L. Zhang, and Y. Zhong, “Auto-ad: Autonomous hyperspectral anomaly detection network based on fully convolutional autoencoder,” IEEE Transactions on Geoscience and Remote Sensing, vol. 60, pp. 1–14, 2022.

[5] J. Liu, Z. Hou, W. Li, R. Tao, D. Orlando, and H. Li, “Multipixel anomaly detection with unknown patterns for hyperspectral imagery,” IEEE Transactions on Neural Networks and Learning Systems, vol. 33, no. 10, pp. 5557–5567, 2022. 493.

Comment (7)

The parameters of the compared methods should be given.

Response:

Thanks for the comment. In our experiment on the comparison methods, the parameters were all set according to the parameters given in their original papers.

Comment (8)

Please include the computation time for all the methods to fully evaluate the proposed methods.

Response:

We greatly appreciate your valuable comment. As shown in Table I, we measured the computing time of our RANet and other comparison methods. Each method was run five times on each data set, taking the average time. As can be seen, although our method shows the best performance, the computation time is relatively high among the ten methods. Reducing computational complexity while maintaining the performance of our method is also our future work.

Table I Computing Time for All the Methods

RANet

RX

LRASR

LSMAD

LSDM-MoG

TC-1

47.25

0.75

44.02

15.85

35.22

TC-2

43.33

0.73

49.34

16.13

54.26

LA

55.14

0.82

45.09

16.66

41.14

SD

52.89

0.67

51.16

19.58

33.86

PTA

SSDF

PAB-DC

Auto-AD

2S-GLRT

TC-1

35.47

35.05

25.52

28.55

30.83

TC-2

37.08

26.11

19.66

38.22

35.17

LA

36.63

29.53

22.25

42.14

39.04

SD

35.63

25.00

20.24

46.86

34.52

Comment (9)

On line 350, the number of cubes P is 10?

Response:

Thanks for the comment. It seems a little strange, but the number of cubes is 100. Thank you anyway for pointing out the possible typo in our manuscript.

Comment (10)

Please have the manuscript thoroughly and carefully checked before submitting the manuscript. I have found several mistakes in this version.

Response:

Thank you for carefully reading our manuscript and pointing this out, we have carefully checked the manuscript and corrected the error.

Special thanks for providing us with your valuable comments, and they have been instrumental in improving our paper. We sincerely hope that our response will meet your expectations.

Reviewer 2 Report

Comments and Suggestions for Authors

The paper proposes an unsupervised method for hyperspectral image anomaly detection based on convolutional autoencoders (CAE). The main contribution of the method is that it combines topological similarity measures computed via a graph attention network with unsupervised CAE to improve its anomaly detection capabilities. CAE reconstruction error is used to detect anomalies. Experiments were carried out on four different real datasets, demonstrating that the proposed method outperforms other alternatives in the literature.

The manuscript is clearly written, well structured, and introduces relevant research. The references included are appropriate and relevant to the topic. The introduction and related work sections are, in my opinion, sufficiently clear and complete. The article provides a detailed exposition of the proposed method, including equations and figures that facilitate understanding.

The figures are mostly appropriate and easy to understand, although some improvements could be made. Additional information on the experimental setup, such as the hardware used for the experiments or the number of experiment runs performed, would be beneficial to validate the accuracy achieved. It is worth noting that there is no reference to the availability of data or codes for the sake of reproducibility.

The conclusions are adequately summarized and in harmony with the experimental results.

Comments:

- Figure 5: The color representations in this figure should be introduced within the figure's caption or in the accompanying text of the article.

- Figure 6: It would be beneficial to include anomaly detection maps that represent hits and misses in the reference data, accompanied by color coding to distinguish true positives, false positives, false negatives, etc.

- Line 284: I expect some explanation on the selection of these specific methods for comparison. In addition, a brief introduction to the distinctive characteristics of each method should be incorporated.

- Figure 8: I suggest reconfiguring this figure into a 2 by 2 format instead of 1 by 4. The current size of the graphs may not adequately facilitate data visualization.

- Table 1: The notable performance of the traditional RX method, which ranks second in most cases, deserves comment, particularly considering the more sophisticated nature of the alternative methods listed in the table. Additionally, the article should include additional details about the results obtained for each method and image. For example, an explanation must be provided for the value of 0.7162 for PAB-DC in the LA data set.

Author Response

Responses for “RANet: Relationship Attention for Hyperspectral Anomaly

Detection”

Dear reviewer,

Thank you for reviewing our manuscript, and providing us with constructive feedback. We have taken your comments into consideration and made revisions to our manuscript. Additionally, we have provided a response to each comment, which is highlighted in blue in our submitted pdf file. Some of the large paragraph description changes are highlighted in yellow (some minor typos and description changes will not be noted) in our new submission. Here, the Summary of Changes is concluded as follows:

In the “Related Work” section, we agree that the description and introduction of some of the latest methods will help readers to have a more comprehensive and in-depth understanding of the HAD task, so we include some of the latest methods proposed in the past two years.

In the “Detection Performance” section, we have added two newer methods of comparison, namely Auto-AD and 2S-GLRT. In addition to this, we have added a paragraph to briefly describe each comparison method.

For Figure 8, we reached an agreement that a 2 by 2 format is better than a 1 by 4 one, since it is structurally more aesthetically pleasing and easier for the reader to read what is in it. So we changed Figure 8 to a 2 by 2 format to make it look clearer.

Thank you again for your time and attention. We hope that you will be satisfied with our revisions and responses.

Best regards,

The authors

Responses

The paper proposes an unsupervised method for hyperspectral image anomaly detection based on convolutional autoencoders (CAE). The main contribution of the method is that it combines topological similarity measures computed via a graph attention network with unsupervised CAE to improve its anomaly detection capabilities. CAE reconstruction error is used to detect anomalies. Experiments were carried out on four different real datasets, demonstrating that the proposed method outperforms other alternatives in the literature.

The manuscript is clearly written, well structured, and introduces relevant research. The references included are appropriate and relevant to the topic. The introduction and related work sections are, in my opinion, sufficiently clear and complete. The article provides a detailed exposition of the proposed method, including equations and figures that facilitate understanding.

The figures are mostly appropriate and easy to understand, although some improvements could be made. Additional information on the experimental setup, such as the hardware used for the experiments or the number of experiment runs performed, would be beneficial to validate the accuracy achieved. It is worth noting that there is no reference to the availability of data or codes for the sake of reproducibility.

The conclusions are adequately summarized and in harmony with the experimental results.

Response:

Thanks for the reviewer’s positive comments. We sincerely appreciate that the reviewer agrees with the strengths of our paper. As for additional information on the experimental setup, we have included the hardware used in the experiment and the number of runs of the experiment in our new submission. You can find it at the end of the “Implementation Details” section. All hyperspectral datasets used in our experiments are publicly available, and we will publish the code as soon as our submission is accepted.

Comment (1)

- Figure 5: The color representations in this figure should be introduced within the figure's caption or in the accompanying text of the article.

Response:

We apologize for the confusion. The different colors used in Figure 5 are only to distinguish the different intervals of values of  and  and do not have any special meaning.

Comment (2)

- Figure 6: It would be beneficial to include anomaly detection maps that represent hits and misses in the reference data, accompanied by color coding to distinguish true positives, false positives, false negatives, etc.

Response:

Thank you for your comment. In fact, in an anomaly detection task, the output anomaly detection graph usually contains probabilities rather than absolute true positives, false positives, false negatives, etc.

Comment (3)

- Line 284: I expect some explanation on the selection of these specific methods for comparison. In addition, a brief introduction to the distinctive characteristics of each method should be incorporated.

Response:

Sorry for the confusion. The methods we choose for comparison are either classic anomaly detection methods or methods that have been popular in recent years. And we apologize for not making a brief introduction to the distinctive characteristics of each method. We have included a brief description of these methods in the “Detection Performance” section, which you can also see below.

Reed Xiaoli (RX) algorithm [1] is proposed assuming that the background obeys multi-variate Gaussian normal distribution. LRASR[2] is a hyperspectral anomaly detection method based on the existence of mixed pixels, which assumes that background data is located in multiple low-rank subspaces. LSMAD[3] realizes anomaly detection by decomposing hyperspectral data into low-rank background components and sparse anomaly components. SSDF[4] is a forest discriminant method based on subspace selection for hyperspectral anomaly detection. LSDM-MoG[5] is a low-rank sparse decomposition method based on mixed Gaussian model for HAD. PTA[6] proposed a tensor approximation method based on priors. PAB-DC[7] is an HAD method based on low rank and sparse representation strategy.

[1] I. S. Reed, X. Yu, Adaptive multiple-band cfar detection of an optical pattern with unknown spectral distribution, IEEE Trans. Acoust. Speech Signal Process. 38 (10) (1990) 1760–1770. doi:10.1109/29.60107.

[2] Y. Xu, Z. Wu, J. Li, A. Plaza, Z. Wei, Anomaly detection in hyperspectral images based on low-rank and sparse representation, IEEE Trans. Geosci. Remote Sens. 54 (4) (2016) 1990–2000. doi:10.1109/TGRS.2015.2493201.

[3] Y. Zhang, B. Du, L. Zhang, S. Wang, A low-rank and sparse matrix decomposition-based mahalanobis distance method for hyperspectral anomaly detection, IEEE Trans. Geosci. Remote Sens. 54 (3) (2016) 1376–1389. doi:10.1109/TGRS.2015.2479299.

[4] S. Chang, B. Du, L. Zhang, A subspace selection-based discriminative forest method for hyperspectral anomaly detection, IEEE Trans. Geosci. Remote Sens. PP (99) (2020) 1–14.

[5] L. Li, W. Li, Q. Du, R. Tao, Low-rank and sparse decomposition with mixture of gaussian for hyperspectral anomaly detection, IEEE Trans. Cybern. 51 (9) (2021) 4363–4372. doi:10.1109/TCYB.2020.2968750.

[6] L. Li, W. Li, Y. Qu, C. Zhao, R. Tao, Q. Du, Prior-based tensor approximation for anomaly detection in hyperspectral imagery, IEEE Trans. Neural. Netw. Learn. Syst. 33 (3) (2022) 1037–1050. doi:10.1109/TNNLS.2020.3038659.

[7] N. Huyan, X. Zhang, H. Zhou, and L. Jiao, “Hyperspectral anomaly detection via background and potential anomaly dictionaries construction,” IEEE Transactions on Geoscience and Remote Sensing, vol. 57, no. 4, pp. 2263–2276, 2019. 478.

Comment (4)

- Figure 8: I suggest reconfiguring this figure into a 2 by 2 format instead of 1 by 4. The current size of the graphs may not adequately facilitate data visualization.

Response:

Thanks for the valuable comment. We agree that a 2 by 2 format is clearer and more beautiful than the original 1 by 4 format, so we have modified Figure 8 in our latest submission following your comments.

Comment (5)

- Table 1: The notable performance of the traditional RX method, which ranks second in most cases, deserves comment, particularly considering the more sophisticated nature of the alternative methods listed in the table. Additionally, the article should include additional details about the results obtained for each method and image. For example, an explanation must be provided for the value of 0.7162 for PAB-DC in the LA dataset.

Response:

Thanks for the valuable comment. Although RX method shows good AUC scores, the RX does not perform well with a low false alarm rate, as shown in Figure 7, and the performance of the methods needs to be judged in conjunction with the AUC scores along with the ROC curves. We apologize for the lack of clarity in the “Experimental Results” section. As for “an explanation must be provided for the value of 0.7162 for PAB-DC in the LA dataset”, we conducted ten additional experiments and chose the optimal result of 0.9323 instead of the original 0.7162. We apologize again for our negligence.

Special thanks for providing us with your valuable comments, and they have been instrumental in improving our paper. We sincerely hope that our response will meet your expectations.

Round 2

Reviewer 1 Report

Comments and Suggestions for Authors

I am satisfied with the current version.